# Surgical Management of Spinal Disorders in People with Mucopolysaccharidoses

**DOI:** 10.3390/ijms21031171

**Published:** 2020-02-10

**Authors:** Hidetomi Terai, Hiroaki Nakamura

**Affiliations:** Department of Orthopaedic Surgery, Osaka City University Graduate School of Medicine, Asahimachi, Abeno-ku, Osaka 545-8585, Japan

**Keywords:** mucopolysaccharidosis, spine, atlantoaxial instability, cervical, stenosis, thoracolumbar, lumbar, surgery, general anesthesia

## Abstract

Mucopolysaccharidoses (MPS) are a group of inherited, multisystem, lysosomal storage disorders involving specific lysosomal enzyme deficiencies that result in the accumulation of glycosaminoglycans (GAG) secondary to insufficient degradation within cell lysosomes. GAG accumulation affects both primary bone formation and secondary bone growth, resulting in growth impairment. Typical spinal manifestations in MPS are atlantoaxial instability, thoracolumbar kyphosis/scoliosis, and cervical/lumbar spinal canal stenosis. Spinal disorders and their severity depend on the MPS type and may be related to disease activity. Enzyme replacement therapy or hematopoietic stem cell transplantation has advantages regarding soft tissues; however, these therapeutic modalities are not effective for bone or cartilage and MPS-related bone deformity including the spine. Because spinal disorders show the most serious deterioration among patients with MPS, spinal surgeries are required although they are challenging and associated with high anesthesia-related risks. The aim of this review article is to provide the current comprehensive knowledge of representative spinal disease in MPS and its surgical management, including the related pathology, symptoms, and examinations.

## 1. Introduction

The mucopolysaccharidoses (MPS) are a group of inherited lysosomal storage disorders involving specific lysosomal enzyme deficiencies that result in accumulation of glycosaminoglycans (GAGs) because of insufficient degradation within cell lysosomes. Each MPS shows a wide spectrum of clinical manifestations, including the onset of signs/symptoms, according to the different residual enzyme activity [1,2,3].

GAG accumulation in prenatal chondrocytes affects primary bone formation and results in severe skeletal manifestations. This accumulation also affects the secondary ossification centers and epiphyseal cartilage growth plates and disturbs the normal systemic endochondral and membranous bone growth after birth, resulting in growth impairment. Once skeletal deformity is established, it is almost irreversible without surgical intervention [4,5]. Features such as dysostosis multiplex, skeletal dysplasia, and resulting short stature suggest the diagnosis of MPS [6,7,8]. The accumulation of GAGs in the ligaments and capsule around the joint causes chronic joint stiffness with limited range of motion (in MPS I, II, III, VI, and VII) or joint hypermobility with ligamentous laxity (in MPS IV) [9,10,11,12,13,14].

Typical spinal manifestations in MPS are atlantoaxial instability (with odontoid hypoplasia), thoracolumbar kyphosis/scoliosis (gibbus deformity), and cervical/lumbar developmental spinal canal stenosis. Spinal disorders and their severity depend on the MPS type and may be related to disease activity. Atlantoaxial instability is frequently observed in MPS IV followed by MPS VI and MPS I. Thoracolumbar kyphosis is a well-known hallmark of MPS I but is also common in MPS II, IV, and VI. Cervical stenosis is broadly recognized in all MPS types except for MPS III [15,16]. (Figure 1).

Enzyme replacement therapy (ERT) or hematopoietic stem cell transplantation (HSCT) has advantages regarding organs such as the liver, skin, and other soft tissue. However, these therapeutic modalities are not effective in MPS-related bone deformity, including of the spine, because of their poor perfusion into bone and cartilage. Thus, bone deformity slowly advances with age, and it is an unsolved problem regarding ERT/HSCT [17,18]. Because spinal disorders show the most serious deterioration among patients with MPS, spinal surgery may be required, although surgery is typically invasive and associated with high anesthesia-related risks [5].

The aim of this review article is to provide the current comprehensive knowledge of representative spinal disease in MPS and its surgical management including the related pathology, symptoms, and examinations.

## 2. Results

### 2.1. Clinical Manifestations and the Surgical Treatment

#### 2.1.1. Atlantoaxial Instability

Atlantoaxial instability (sometimes termed “occipito–cervical instability”) is a common pathology in MPS IV and VI and is accompanied by hypoplasia of the dens and the C1 posterior arch, ligamentous laxity, and extradural accumulation of GAGs [19,20,21]. Impingement between C2 and the C1 posterior arch in flexion/extension may cause irreversible spinal cord damage resulting in sudden death or in respiratory failure with quadriparesis [22]. Neurological signs and symptoms vary. Numbness and clumsiness in patients’ hands, hyperreflexive deep tendons, hypotonia of the extremities, and general fatigue are single or combined signs of upper cervical cord compression. However, MPS is sometimes difficult to diagnose correctly because of communication difficulties related to the patient’s age or developmental delay and joint contracture masking the underlying disease. The earliest complaint of patients with upper cervical compression is sometimes loss of endurance, diminished walking distance, and gait instability [23]. Motor signs tend to be more severe in the lower extremities, but the upper extremities may be affected as well [24].

Atlantoaxial instability is usually evaluated on flexion/extension radiographs by measuring the distance between the posterior wall of the odontoid process and anterior wall of the C1 posterior arch. However, in cases of odontoid hypoplasia, the cranial edge of the odontoid process is substituted for the odontoid process in plain radiographs and termed “the space available for the cord” [25]. The space available for the cord is normally wider at the atlantoaxial level than that at subaxial levels [26], although this level is usually the narrowest in patients with MPS with atlantoaxial instability. Assessing hypoplasia of the odontoid is more accurate in sagittal computed tomographic (CT) images than in magnetic resonance images (MRI) or plain radiographs because the lack of ossification is easily identified. A compressed spinal cord can be clearly evaluated by MRI, although its incidence can be underestimated because MRI is usually performed in the neutral position. Therefore, flexion/extension MRI using an open system is useful to evaluate the degree of compression, if this imaging option is available [27]. High signal intensity within the spinal cord in T2-weighted images at the compressive level is a sign of edema or myelomalacia, which suggests irreversible damage to spinal cord neurons. However, clinical symptoms are not always correlated with MRI changes [28]. (Figure 2) Giussani et al. described the importance of early recognition of cranio-vertebral joint to avoid severe complications [29].

Regarding the timing of operation, it is most important to perform fixation/decompression surgery to relieve instability and compression before irreversible cord damage occurs. However, it remains unclear when to perform early prophylactic fusion surgery to prevent future spinal cord damage. Furthermore, there are problems associated with this surgery in patients with MPS. First, it is difficult to use instruments for fixation such as plates and screws for immature small vertebrae, and it is also difficult to maintain stability using orthoses, postoperatively. The development of cervical vertebrae and bony union between the vertebral body and the laminae usually occurs by 3–6 years of age, at which time the size of the spinal canal almost matches that of adults [30]. Therefore, fusion surgery should be performed after the maturation of the cervical vertebrae if the patient’s neurological state allows. Subaxial adjacent instability requiring additional surgery is another problem with early spinal fusion. Ozgur et al. reported that 7/20 (35%) patients with MPS IVa required revision surgery 36–168 months after their initial surgery because of adjacent level problems [21]. Mollmann et al. reported the efficacy of a scoring system for assessing the severity of cervical cord problems in patients with MPS IVa to determine the optimal surgical timing [28].

Several operative methods have been introduced for atlantoaxial/occipito–cervical fixation in patients with MPS, namely, in situ fusion using a halo-vest, cables or wires, transarticular screws, or laminar screws [21,31,32,33]. The surgical technique is selected depending on the patient’s age and anatomical vertebral size, and the surgeon’s preference. If there is enough atlas antero-posterior diameter and size, atlantoaxial fixation is preferred to preserve occipito–atlanto motion. However, occipito–cervical fusion should be selected when C1 laminectomy is required to decompress the spinal cord because of a hypoplastic or immature C1 posterior arch. Vertebral arteries and related anatomy must be evaluated by contrast-enhanced CT before surgery to avoid intraoperative vascular injuries, and an intraoperative navigation system is recommended, if available [34]. There is no clear evidence regarding bone union rates after bone grafting spinal surgery, but the rate appears to be similar to that in normal conditions and depends more on the surgical technique than on osteogenic ability in MPS [35]. (Figure 3) Krenzlin et al. reported the feasibility of posterior arch resection alone in 15 patients with MPS with craniocervical stenosis; however, the authors did not consider atlantoaxial instability, and nearly half of the patients required revision surgery [36]. It is unclear whether decompression of C1 alone accelerates future instability, and distal junctional instability is a major problem after occipito–cervical fusion. Upper cervical fusion provides reliable neural outcomes [21], and laminectomy of C1 alone in young patients should be avoided because this carries the risk of atlas fracture [37].

#### 2.1.2. Cervical Stenosis

Cervical stenosis is widely recognized in MPS I, II, VI, VII, and mucolipidosis [16] and is the main spinal problem in patients with MPS [38]. There are two pathophysiological factors in spinal stenosis in MPS: developmental and acquired factors. Developmental spinal canal stenosis is classified into two types: hereditary–idiopathic stenosis and skeletal growth disorder-related stenosis-like MPS [39]. Acquired factors are related to GAG accumulation in the connective tissues surrounding the epidural space, especially in the subligamentous membrane between the dura and the ligamentum flavum, a hypertrophied ligamentum flavum, degenerated facet joints, and intervertebral discs. GAG accumulation is mainly observed between the dura and ligamentum flavum and not within the dura mater or subarachnoid space; therefore, it is important to remove these deposits during surgery to achieve complete decompression.

The symptoms of cervical stenosis depend on the severity and compressed level of the spinal cord. C1 hypoplasia with atlantoaxial instability is a common pathology in MPS IV and VI; however, this is not common in other MPS types although hypoplasia of the C1 posterior arch is widely recognized. The main cause of cord compression at the C1 level is GAG accumulation behind the dens combined with hypoplasia of the C1 posterior arch. In most patients, the initial symptom of cord compression is numbness in both hands. A neck extension test (considered positive if a patient feels numbness in both hands when they extend their neck), Phalen’s test, and Tinel’s sign are necessary for diagnosis because carpal tunnel syndrome is also common in MPS I, II, and III [15,40,41].

MRI is required to diagnose cervical stenosis. A thickened ligamentum flavum and hypertrophic changes in the circumferential epidural space caused by GAG accumulation and spinal canal narrowing with obstructed cerebrospinal fluid flow are evidence of a compressed spinal cord. Progressive degenerative changes in the intervertebral discs are also observed in patients with MPS compared with patients of the same age with a normal cervical spine. If intramedullary high-intensity changes in T2-weighed images are recognized, irreversible cord damage is suspected [19] (Figure 4). Additionally, the spinal canal appears as a small triangular shape with thickened laminae compared with a normal cord, which looks wider with CT (Figure 5). The width of the spinal canal measured in lateral plain radiographs should be > 13 mm in adult men and 12 mm in adult women, and is defined as developmental canal stenosis if these measurements are smaller [42,43]. This definition of developmental canal stenosis is not useful for patients with MPS because vertebral size is originally smaller, and the spinal canal is narrower. Therefore, narrowing of the cervical canal should be evaluated by comparisons with unaffected spinal cord levels.

Patients with cervical stenosis should undergo decompressive surgery before irreversible cord damage occurs [19]. Posterior decompression surgery without instrumented fixation is appropriate for patients without obvious cervical instability. Laminoplasty using artificial hydroxyapatite spacers is preferable to avoid complications, further instability, and post-laminectomy membrane fibrosis compared with total laminectomy [44,45]. In most cases, the addition of C1 laminectomy is recommended combined with C2–C7 laminoplasty, and the necessity of suboccipital decompression should be considered when stenosis at the occipito–cervical is suspected (Figure 6). In young patients with immature cervical laminae, total laminectomy is the only surgical option. In these patients, recurrence of stenosis secondary to instability or laminectomy membrane fibrosis should be carefully monitored long-term. Mayfield fixators are used for adult patients with MPS to stabilize the head and neck during surgery, but a halo-vest is necessary for infants with MPS (Figure 7). Kawaguchi et al. investigated patients who had undergone laminoplasty more than 20 years previously and found that 49.2% of the survivors had worsened clinical outcomes measured by the Japanese Orthopedic Association score during follow-up, and the patients in the worsened group had undergone additional spinal operations [46]. There are currently no long-term follow-up data for patients with MPS undergoing cervical laminoplasty or laminectomy. However, findings similar to those of Kawaguchi et al. are expected because of the extended life expectancy in patients with MPS.

#### 2.1.3. Thoracolumbar Kyphosis

Thoracolumbar kyphosis is caused by deformity of the vertebral column at the thoracolumbar junction and progresses with age [6]. Thoracolumbar kyphosis occurs in MPS I, II, IV, and VI, while scoliosis can occur in conjunction with kyphosis or in isolation in MPS I, II, and III [47]. The specific shape of the vertebra in MPS is called platyspondyly, which is secondary to incomplete endochondral ossification. Coined vertebra or beaked vertebra are terms that also represent the characteristic shape of the vertebral deformity in MPS [48]. Anterior beaking of the inferior aspect of the vertebral body is common in MPS I, while patients with MPS IV show anterior beaking of the midpoint of the vertebral body. However, it is still difficult to determine the type of MPS only by the shape of the vertebral deformity [49]. Thoracolumbar kyphosis causes sagittal imbalance accompanied by thoracic lordosis, pelvic anteversion, and lumbar hyper-lordosis, requiring flexed knee and hip joints to maintain standing balance [50].

Regarding the treatment of thoracolumbar kyphosis in MPS, there is no clear evidence for the ideal timing of operation, the ideal surgical method, or the efficacy of bracing. Studies report that thoracolumbar kyphosis exceeding a 40-degree Cobb angle has the propensity to progress in MPS I [51]. The effect of ERT or bone marrow transplantation on preventing spinal deformity is not confirmed and requires further examination of the natural history of kyphosis progression in MPS [24]. Abelin et al. suggested using bracing to prevent kyphosis progression immediately after patients are able to maintain the sitting position [50], and Crostelli et al. reported successful outcomes using braces in 15 patients with MPS I [49].

Progressive kyphosis with neurological deficit is commonly accepted by spine surgeons as a definitive indication for surgical intervention. Williams reported in a literature review of 58 patients with MPS (MPS I: 38 patients, MPS II: 3 patients, MPS IV: 9 patients, MPS VI: 8 patients), that perioperative neurological compromise associated with thoracolumbar kyphosis was observed only in MPS IV and VI (MPS IV: 5 patients, MPS VI: 2 patients). According to these results, the surgical indications for MPS I should be considered carefully because there are no long-term data describing the natural history of the progression curve as in other skeletal dysplasias, and there is also a lack of functional and quality of life assessments after thoracolumbar surgery [52]. Postoperative complications include death, neurological deficits, pseudarthrosis, infection, additional junctional deformity, and anesthesia-related problems. Additionally, perioperative thoracic spinal cord ischemia is a suspected cause of paraplegia [53]. We experienced cardiac arrest in a patient with MPS I during kyphoscoliosis surgery and had to discontinue the surgery after inserting the pedicle screws [54]. The patient was transferred to the intensive care unit and recovered completely after pacemaker insertion. The patient showed no progression of kyphoscoliosis 10 years postoperatively, but experienced moderate low-back pain. Surgeons should base the need for surgery not only on radiographical changes but also on the patient’s neurological function and quality of life (QoL), especially in patients with MPS I (Figure 8).

#### 2.1.4. Lumbar Canal Stenosis

Lumbar spinal canal stenosis (LCS) is a common disease among populations older than 60 years of age. MPS is a known cause of LCS and has been categorized as congenital/developmental stenosis since the 70’s [55]. However, LCS has received less attention than cervical stenosis or thoracolumbar kyphosis because the symptoms of LCS are not critical, and patients had more severe systemic problems to address before the introduction of ERT or HSCT. Hypertrophy of the ligamentum flavum, degenerative changes in the facet joints, and bulging intervertebral discs are the three main factors causing degenerative LCS. In addition to these factors, developmental canal stenosis related to bone deformity may largely explain LCS in patients with MPS. Bone deformity in lumbar lesions includes posterior scalloping and anterior beaking caused by GAG accumulation in the superior and inferior cartilaginous plates, upon which vertebral body growth depends [56].

The main symptoms of LCS are intermittent claudication, low-back pain, and bowel and/or bladder dysfunction. Intermittent claudication is remarkable only in normal ambulatory patients with MPS who can walk a long time without assistance. Therefore, most LCS patients have the attenuated type of MPS I. As in non-MPS patients, degenerative changes such as disc bulging and hypertrophic changes in the ligamentum flavum have important roles in developing LCS in patients with MPS. Spondylolisthesis, hypoplastic facets, deformed endplates, and vertebral malformation, which are commonly observed in MPS, accelerate degeneration. Thus, younger patients with MPS are more likely to develop LCS compared with patients without MPS. As life expectancy increases with ERT/bone marrow transplantation therapies, degenerative orthopedic problems such as LCS will become more evident [57,58].

The surgical indications for LCS in patients with MPS are the same as for the non-MPS population. Posterior decompression of involved levels alone is enough when there is no remarkable spinal instability. Instrumented fusion is only indicated if there is instability with spondylolisthesis or isthmic spondylolysis affecting patients’ symptoms. No long-term follow-up data are available regarding LCS surgery in patients with MPS (Figure 9).

#### 2.1.5. General Anesthesia for Spinal Surgery

Spinal surgery for patients with MPS is challenging. Hypertrophied tonsils and epiglottis, macroglossia, and abundant salivary secretions in MPS are known risk factors during intubation and extubation in general anesthesia. In most spinal operations, patients must remain in the prone position for a long time, which leads to difficulty controlling the airway. Increased abdominal pressure in the prone position is another problem, however, it is necessary to relieve abdominal/airway pressure to minimize intraoperative bleeding [59]. Because of short stature/necks and abdominal bloating in patient with MPS, the prone position is difficult but necessary to perform surgery. Careful control of infusions and blood transfusions is required to maintain patients’ circulatory dynamics during surgery. Excess infusion easily increases cardiac afterload and decreases cardiac reserve. Arn et al. reported that 32/753 (4.2%) patients with MPS I who had undergone at least one surgery died within 1 month of a surgery. Most deaths were not related to surgery but rather perioperative pulmonary or cardiac complications [60]. Patients must be evaluated for cardiac, pulmonary, and airway conditions by specialists before surgery to avoid perioperative risks. Anesthesiologists should evaluate cervical MRI/CT images to determine the direction of neck movement that does not impact the cervical cord during intubation [38,61,62].

## 3. Conclusions

Establishing appropriate surgical indications for spinal disorders in MPS remains a challenge though guidelines of MPS have recently been launched [63,64]. Despite new therapeutic modalities such as ERT/HSCT, which are expected to contribute prolonging the life expectancy of patients with MPS, problems remain regarding musculoskeletal disorders, especially spinal problems. Cervical lesions are more critical when considering patients’ QoL. Surgeons must be aware of typical symptoms and understand the underlying pathology to avoid delaying surgical interventions. Regarding thoracolumbar and lumbar lesions, long-term follow-up data are not available, including for the natural history, and there are no data defining how spinal surgeries contribute to improved QoL because of the rarity of these lesions. Degenerative processes and related musculoskeletal disease will be more important for MPS patients than for non-MPS populations because of the successful introduction of ERT/HSCT. Collecting more clinical data and further study are required to determine the necessity and optimal timing of surgery and the best surgical approach for deformity and lumbar lesions.

## Figures and Tables

**Figure 1 ijms-21-01171-f001:**
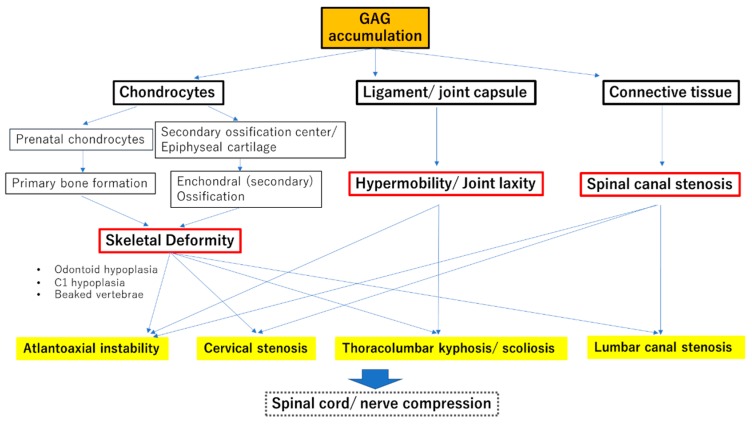
Pathophysiology of spinal cord/ nerve compression in mucopolysaccharidoses (MPS). Pathophysiology of all spinal problems in MPS are accountable by combination factors of skeletal deformity, hypermobility/ joint laxity, and spinal canal stenosis. Skeletal deformity including odontoid hypoplasia, C1 hypoplasia and other vertebral deformities is caused by glycosaminoglycan (GAG) accumulation in chondrocytes of primary/secondary ossification centers. It starts in prenatal period and continues until bone growth stops. Every spinal problem in MPS is based on skeletal deformities. GAG accumulation in spinal ligament and capsule of facet joints, inducing inflammation and degradation of surrounding tissue, can result in hypermobility and joint laxity, whereas that occurred in connective tissue in anterior extradural space and ligamentum flavum leads to spinal canal stenosis. Atlantoaxial instability is caused by skeletal deformity, hypermobility, and spinal canal stenosis, though the lack of instability leads to cervical stenosis. Skeletal deformity is the first and fundamental factor of all spinal problems. Thus, early diagnosis and intervention is necessary to avoid skeletal deformity and further deterioration of spinal cord/nerve.

**Figure 2 ijms-21-01171-f002:**
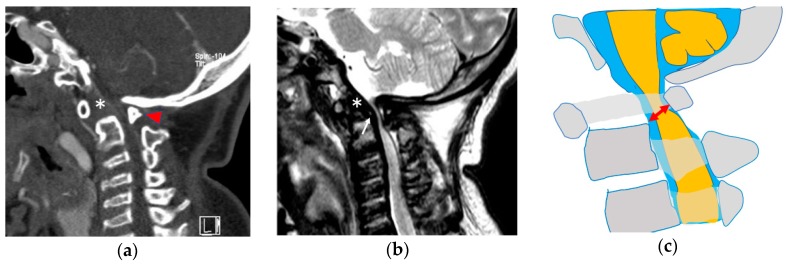
Atlantoaxial instability in MPS IVa. Hypoplasia of the dens (asterisk) and C1 posterior arch (arrowhead) are more clearly recognized in computed tomography (**a**), while GAG accumulation (white arrow) and spinal cord compression are more identifiable in magnetic resonance images (**b**). The space available for the cord (red arrow) indicates the most narrowing between the posterior wall of C2 and the C1 posterior arch (**c**). This patient had tetraplegia and required ventilator control because of irreversible spinal cord damage at the craniocervical junction.

**Figure 3 ijms-21-01171-f003:**
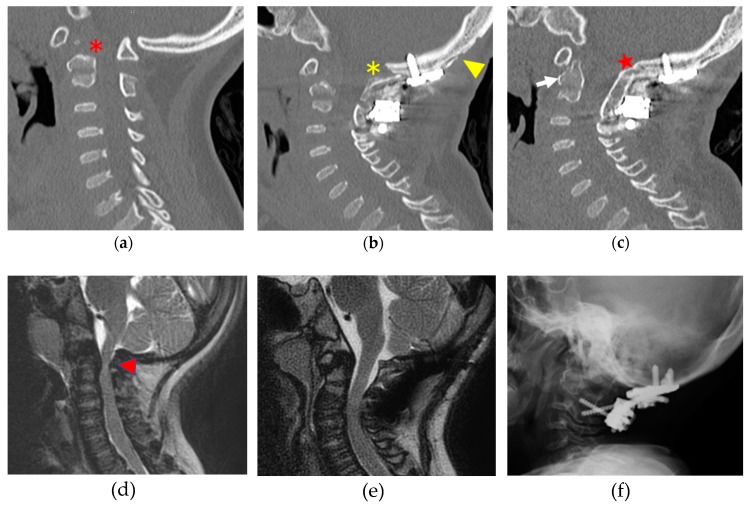
Occipito–cervical fusion in MPS IVa (10-year-old girl). Hypoplasia of the dens and the C1 posterior arch (asterisk) in the computed tomographic image causing spinal cord compression and myelomalacia (arrowhead) in the magnetic resonance image before surgery (**a**,**d**). Occipito–C2 fusion surgery was performed, and the spinal canal was totally decompressed at the occipito–cervical junction. The outer plate of the occipital bone (yellow arrowhead) was used for bone grafting material (yellow asterisk) (**b**,**e**). Good bone union (red star) was achieved and, interestingly, bone formation in the dens (white arrow) was observed 10 months postoperatively. C2 laminar screws combined with occipital plating were used for this surgery (**c**,**f**).

**Figure 4 ijms-21-01171-f004:**
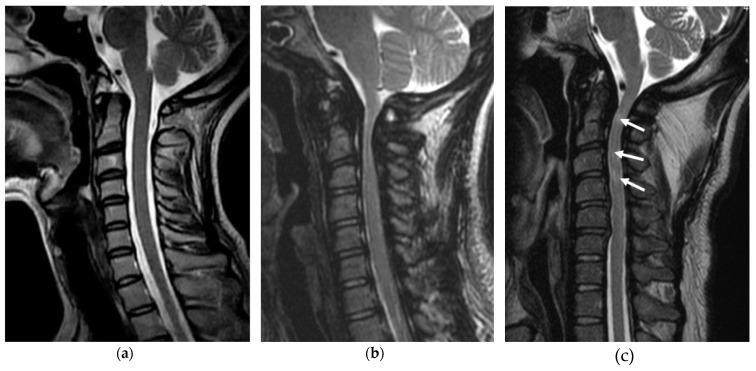
Cervical stenosis in MPS. Typical sagittal T2-weighed magnetic resonance image of the cervical spine in a normal adult (**a**), MPS I (31-year-old woman) (**b**), and MPS II (37-year-old man). GAG accumulation and stenosis is evident, especially around the odontoid process. Multiple T2 high-signal-intensity changes are observed in the spinal cord (arrows). Compared with the upper thoracic levels, blocked cerebrospinal fluid flow secondary to the hypertrophied ligamentum flavum is seen (**c**).

**Figure 5 ijms-21-01171-f005:**
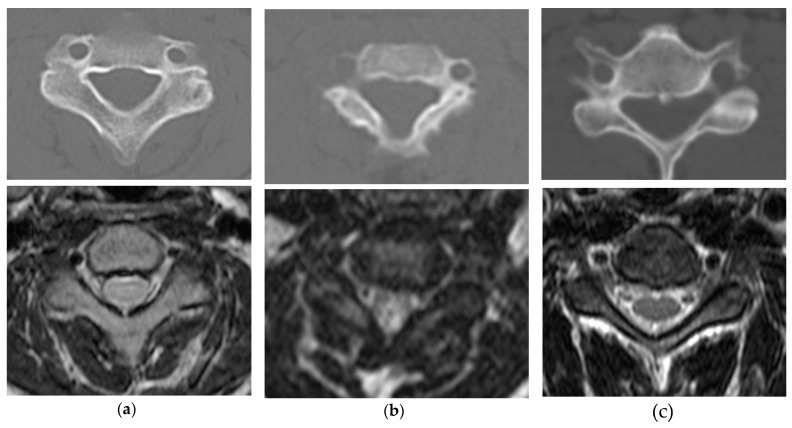
Shape of the spinal canal and spinal cord in MPS. 13-year-old boy with MPS II (**a**) and 3-year-old boy (**b**). The bony spinal canal is composed of thickened laminae and small vertebrae with less cerebral spinal fluid compared with a 48-year-old normal man (**c**). High-signal-intensity change in T2-weighed magnetic resonance image is seen (**b**). The upper column (computed tomographic images) and the lower column (T2-weighed magnetic resonance images) represent the same cervical level.

**Figure 6 ijms-21-01171-f006:**
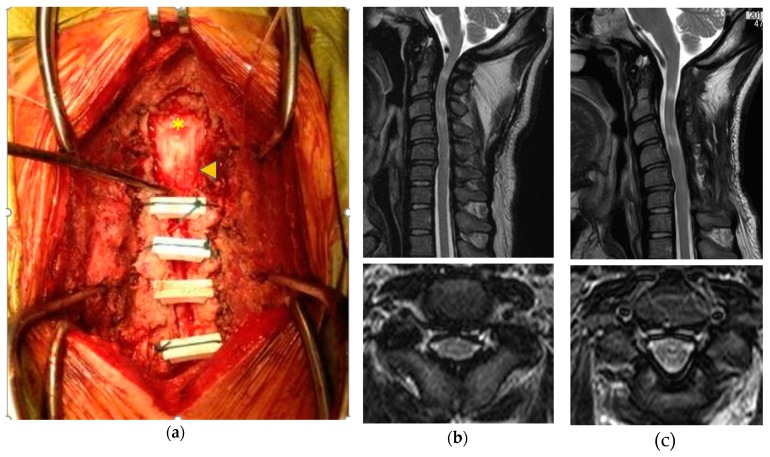
Cervical decompression surgery for MPS. Open-door laminoplasty from C3–C6 using hydroxyapatite spacers combined with C1 and C2 laminectomy (arrowhead) and suboccipital decompression (yellow asterisk) was performed in a 37-year-old man with MPS II (**a**). Cervical stenosis before surgery (**b**) was totally decompressed postoperatively (**c**). Numbness and clumsiness in both hands improved, although high signal intensity was still present after surgery. (The upper images are sagittal and the lower ones are axial T2-weighed magnetic resonance images in (**b**,**c**)).

**Figure 7 ijms-21-01171-f007:**
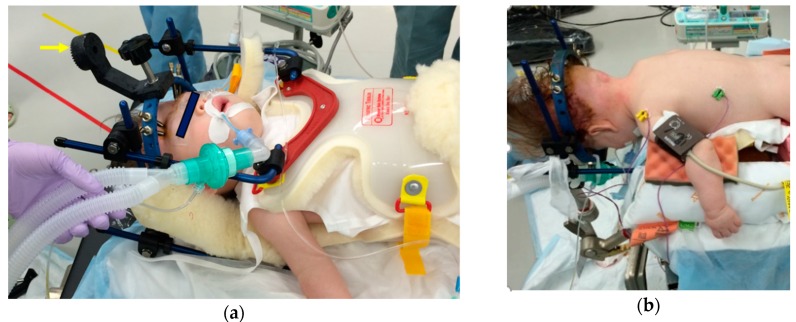
Patient positioning for an infant with MPS using a halo-vest. The halo-vest was placed under general anesthesia before surgery in the supine position (**a**) and connected to the operating table via a special connector (arrow in image (**a**)) in the prone position (**b**). We used the halo-vest to stabilize the patient’s neck, instead of a neck orthosis, for 2–3 weeks postoperatively.

**Figure 8 ijms-21-01171-f008:**
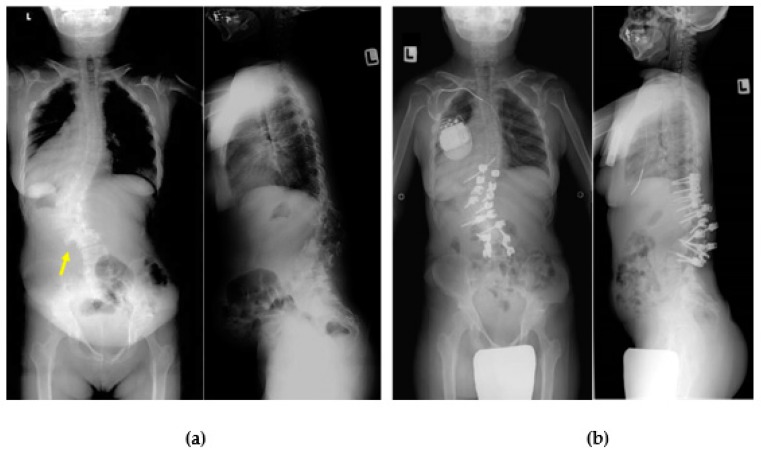
Thoracolumbar kyphosis in attenuated MPS I. X-ray images of a 31-year-old woman with MPS I with low-back pain showing kyphoscoliosis and lateral slip of the lumbar spine (arrow) (**a**). Cardiac arrest occurred during surgery in the prone position, which required discontinuing surgery just after completing the pedicle screw insertion. The patient recovered fully without critical damage, although she required a pacemaker (**b**). More than 10 years postoperatively, the degree of kyphoscoliosis and low-back pain had not worsened.

**Figure 9 ijms-21-01171-f009:**
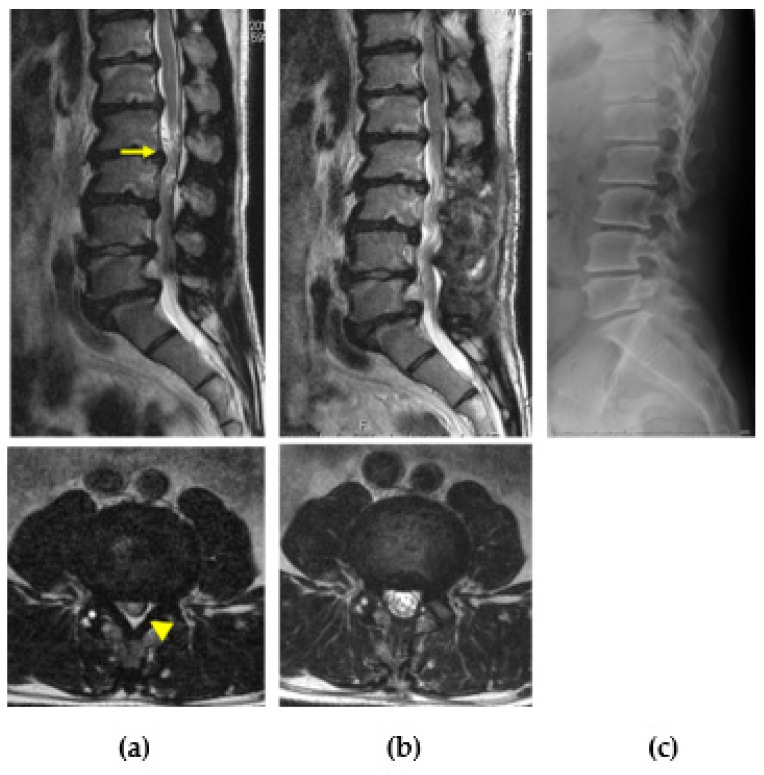
Lumbar canal stenosis in MPS II. Preoperative magnetic resonance images of a 37-year-old man with MPS II showing lumbar canal stenosis with redundant nerves (arrow), multiple disc degeneration and bulged discs, and hypertrophied ligamentum flavum (arrowhead) (**a**). Intermittent claudication disappeared after posterior decompression surgery. The spinal canal was enlarged, and nerve root redundancy also disappeared in postoperative magnetic resonance images (**b**). Vertebral deformity was not evident compared with MPS IV; however, isthmic spondylolysis and degenerative changes were observed in the plain radiograph (**c**).

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
