# Peer review of "Surgical Management of Spinal Disorders in People with Mucopolysaccharidoses"

_ijms, 2020, doi:10.3390/ijms21031171_

Round 1

Reviewer 1 Report

In this paper the authors performed a pictorial review on the spinal disorders linked to MPS. They synthetically treated pathophysiology, clinical features and surgical treatments for this disease complex. All in all the topic is interesting and the paper is pretty well written, but it partially goes beyond the field of interest of most of the readers of the IJMS. For this journal, they should have done a more extensive description to the physiopatological elements rather than a brief description of the surgical technique. On the other hand, the discussion about surgical treatment appears do not add anything new to the current literature for a technical orthopedic journal.

Author Response

 Thank you for your review and helpful comments. We have added the pathophysiological element how spinal cord compression occurs in MPS as figure 1. We think three factors skeletal deformity, ligament laxity and spinal canal stenosis are main composers of spinal problems in MPS. Among them, skeletal deformity is the one molecular biologists could target on.

Regarding surgical treatments, we have listed and described latest articles about spinal surgery of mucopolysaccharidosis which are available through PubMed. We tried to find new one but could not find.

Reviewer 2 Report

This review paper provides a summary of spinal disorders in MPS and methods of their surgical treatment. This contribution is scientifically sound, however, presnetation style and English usage must be considerably improved. There are many English errors, starting from the first sentence in the Abstract ("Mucopolysaccharidoses (MPS) are a groups" - a mixture of singular and plular, gramatically incorrect), but there are far too many of them to list even a small part. Therefore, the text must be corrected by a native English speaker or somebody professional in English. Because of the language problems, some sentences are even completely unclear (e.g. page 3, lines 16-17: "It is not unclear decompression surgery alone of C1 will accelerate future instability..." - this does not make sense, but this is a language problem rather than incorrect merit). Moreover, in the list of references, there are double numbers of cited articles, and from ref. 13, the numbers are inconsistent. This should be corrected.

Author Response

Thank you for your review and comments. We have sent the draft to the professional editing service. (Jane Charbonneau, DVM, from Edanz Group (www.edanzediting.com/ac)) and changed including title. I believe the English writing now meets the standard. Regarding references, I corrected the numbers.

Reviewer 3 Report

Dear Authors,

congratulations for the great work and for the aim of the paper to face a very important and critical point in the management of MPS patients.

In the abstract I suggest to consider the multisistemic involvement due mainly to the GAGs storage in MPS. GAGs are ubiquitous proteoglycans. 

In the page 2 line 2 I suggest to specify as HSCT is generally performed for brain: in fact ERT do not pass blood brain barrier and so in neurological forms of MPS the purpose of HSCT is to preserve central nervous system.

At page 2 line 41 I think 'if' is omitted.

Good discussione about atlantoaxil instability, cervical stenosis and thoracolumbar kyphosis with many details, I appreciate a lot.

At page 4 line 49 you referred to MPSI but I suggest to specify who is a Scheie patients (with mild form).

The discussion in thoracolumbar kyphosisi is too articulated.

At page 5 line 26 macroglossia is to be added as complication of intubation.

An upper airways study is often needed before anhestesia for the risk of tracheal stenosis. 

The cardiological and respiratory complications are also very important factors that need evaluation to avoid risk.

In the conclusion you may mention the recent guidelines of MPS IV and VI:

Akyol-MU et al, Orphanet J Rare Dis 2019

At page 11 line 3  'ERT/HSCT that prolonged life expectacy'. We need multicenter collaborative studies and rigorous strumental anc clinical management of these patients.

An interesting paper would be cited: Giussani C et al, Italian J Pediatr 2018;44:119.

Author Response

Thank you for review and valuable comments.

We have sent the manuscripts to professional editing service to brush up texts.

In the abstract I suggest to consider the multisistemic involvement due mainly to the GAGs storage in MPS. GAGs are ubiquitous proteoglycans.

Thank you for your valuable comments. I understand GAGs are ubiquitous proteoglycans and MPS is multisystem disease. I have added “multisystem” in 1st sentence in the abstract.

In the page 2 line 2 I suggest to specify as HSCT is generally performed for brain: in fact ERT do not pass blood brain barrier and so in neurological forms of MPS the purpose of HSCT is to preserve central nervous system.

Thank you for the valuable comments. I discussed ERT/ HSCT without differentiation because I described the effects of both ERT and HSCT only on bone. I did not discuss about the central nervous system because of I focused on spinal problems caused by skeletal deformity in this review.

At page 2 line 41 I think 'if' is omitted.

I changed the sentence.

At page 4 line 49 you referred to MPSI but I suggest to specify who is a Scheie patients (with mild form).

Sorry for bothering. I removed the word “Scheie” and unified “MPS I”.

At page 5 line 26 macroglossia is to be added as complication of intubation.

Thank you for your valuable comments. I have added “macroglossia” as complication.

An upper airways study is often needed before anhestesia for the risk of tracheal stenosis.

The cardiological and respiratory complications are also very important factors that need evaluation to avoid risk.

Those factors should be checked before surgery. I described in the text” Patients must be evaluated for cardiac, pulmonary, and airway conditions by specialists before surgery to avoid perioperative risks. “

In the conclusion you may mention the recent guidelines of MPS IV and VI:

I have added the sentence in conclusion and references.

At page 11 line 3  'ERT/HSCT that prolonged life expectacy'. We need multicenter collaborative studies and rigorous strumental anc clinical management of these patients.

We have changed sentence as following, “Despite new therapeutic modalities such as ERT/HSCT, which are expected to contribute prolonging the life expectancy of patients with MPS, problems remain regarding musculoskeletal disorders, especially spinal problems.”

An interesting paper would be cited: Giussani C et al, Italian J Pediatr 2018;44:119

Thank you for the suggestion. I have added the sentence and reference following, “ Giussani et al. described the importance of early recognition of cranio-vertebral joint to avoid severe complications.[29]”

Round 2

Reviewer 1 Report

The authors did their best to improve the manuscript according to the suggestion of the reviewers.